# An Approach to Measure Tilt Motion, Straightness and Position of Precision Linear Stage with a 3D Sinusoidal-Groove Linear Reflective Grating and Triangular Wave-Based Subdivision Method

**DOI:** 10.3390/s19122816

**Published:** 2019-06-24

**Authors:** Hsiu-An Tsai, Yu-Lung Lo

**Affiliations:** 1Department of Mechanical Engineering, National Cheng Kung University, Tainan 70101, Taiwan; n18991152@mail.ncku.edu.tw; 2Department of Industrial Upgrading Service, Metal Industries Research & Development Centre, Kaohsiung 81160, Taiwan

**Keywords:** detectors, metrological instrumentation, optical measurement

## Abstract

This work presents a novel and compact method for simultaneously measuring errors in linear displacement and vertical straightness of a moving linear air-bearing stage using 3D sinusoidal-groove linear reflective grating and a novel triangular wave-based sequence signal analysis method. The new scheme is distinct from the previous studies as it considers two signals to analyze linear displacement and vertical straightness. In addition, the tilt motion of the precision linear stage could also be measured using the 3D sinusoidal-groove linear reflective grating. The proposed system is similar to a linear encoder and can make online measurements of stage errors to analyze automatic processes and also be used for real-time monitoring. The performance of the proposed method and its reliability have been verified by experiments. The experiments show that the maximum error of measured tilt angle, linear displacement, and vertical straightness error is less than 0.058°, 0.239 μm, and 0.188 μm, respectively. The maximum repeatability error on measurement of tilt angle, linear displacement, and vertical straightness error is less than ±0.189^o^, ±0.093 μm, and ±0.016 μm, respectively. The proposed system is suitable for error compensation in the multi-axis system and finds application in most industries.

## 1. Introduction

The demand for ultra-precision technology, high-precision multi-degrees-of-freedom displacement measurements for industrial manufacturing and inspection applications is increasing steadily. Physical parameters like linear and angular displacements play a key role in improving the quality of a production system. In addition, high-precision stages are commonly used for semiconductor processing, PCB (Printed Circuit Board) drilling, micromachining, precision assembly, and inspection processing. Although measuring and specifying static/quasi-static straightness is a well-established process in existing performance standards [1,2], a standard test for characterizing dynamic straightness of single-axis linear positioning systems has not yet been developed. This is the reason for many new ultra-precision linear positioning systems finding their way into emerging technologies that require exceptional straightness during both static/quasi-static and dynamic positioning. As a result, manufacturers and users of linear positioning systems follow their own methods and standards creating confusion among customers.

Geometric errors, such as straightness, flatness, yaw, pitch, and roll errors [3], inevitably limit the performance of linear stages. Since a motor stage moves along the sliding guide without losing mechanical contact, the making of high-precision linear stages depends crucially on high-precision machining accuracy and assembling skills [4]. Yet, high machining and assembling accuracy is hard to achieve for long-stroke stage. Apart from position measurement, the measurement of the displacement vertical to the direction of motion, commonly known as straightness measurement, constitutes a challenging task for precision stage [5]. Interestingly, the term “straightness” generally refers to a broad spectrum of engineering quality from work-piece straightness to motion straightness. Today, optical means are widely used to measure straightness errors in engineering metrology, in which straightness data are obtained through various optical accessories, such as autocollimators, alignment telescopes and optical theodolites. Typically, to make straightness measurement, it is necessary to deploy a sensor, such as the straightness measurement kit of a laser interferometer or a capacitance-type displacement sensor combined with a straight edge. Consequently, high cost and intrinsic complexity have made optical measurement systems commercially disadvantageous.

Laser interferometer, such as from Agilent Co. or Renishaw Plc, is arguably the most representative instrument for geometric error calibration. It allows direct measurement of geometric errors along each axis independently. Other methods to reach high alignment accuracy, such as polarimetry method [6], polarizing method [7], optical compensation method, and the Zeeman laser interferometer method [8], have also been developed. Among the techniques used for laser interferometric straightness measurement are conventional laser interferometer and laser straightness interferometer, specifically designed to measure straightness errors. Yang et al. [9,10] and Fan et al. [11] used multiple laser interferometers to measure the straightness error of linear motor stage, while Chen et al. [12] developed a straightness interferometer using a non-polarizing beam-splitter and a polarizing beam-splitter to obtain two separate measurement signals, which simultaneously measure the magnitude and the position of straightness errors.

Yet, in either setup, the laser interferometer system can measure only one error element; moreover, the optical setup and laser path alignment before measurement are time-consuming processes. In this regard, new methods and measuring systems have been developed for laser collimation. A position-sensitive detector (PSD) or a quadrant photo diode (QPD) is usually used as a sensor to measure straightness errors. These devices are typically low cost, and for optical adjustment are simple and fast. Chen et al. [13] designed a six-degrees-of-freedom optical sensor for dynamic measurement of linear axes that use a position-sensitive photodiode to measure straightness errors and a heterodyne interferometry to measure linear displacements. In addition, Jywe et al. [14] developed a multi-degrees-of-freedom measuring system and an error-compensation technique for machine tools integrating a laser interferometer for measuring displacements and a QPD for detecting straightness errors. Compared with laser collimation and diffracted beam interference method that uses grating [15,16,17,18] and polarization angle-detecting method [19,20], the advantage of using laser interferometry for straightness measurement is twofold: nanometer-scale accuracy and long travel range. Unfortunately, due to the overall heaviness of the structure, the above measurement system is difficult to install on precise positioning machinery for online measurement.

While commercial optical encoders are commonly used in modern positioning systems, the measurement setup as discussed above to detecting the position and straightness along the x-axis ends up by making it large and complicated. Also, it is necessary to use additional auto-collimators for measuring the tilt motions of the precision stages. Moreover, the accuracy and resolution of an optical encoder are constrained by generated sinusoidal signals and the assembly technique. To overcome these constraints, we present here a novel and compact method, which simultaneously measures linear displacements and vertical straightness errors of a linear moving air-bearing stage. This method is based on a three-dimensional (3D) sinusoidal-groove linear reflective grating structure and a novel triangular wave-based sequence signal analysis method. This paper also has derived analytical equations to extract the tilt motion of the linear stage.

## 2. Overall Scheme and Measurement Principle

Figure 1 shows the schematic of the measurement system. The sensor is stationary and the stainless-steel bar is placed in the x-z plane of the linear stage, which is along the x-axis. The collimated laser beam is incident on the beam-splitting device. The polarizing beam splitter, which is positioned along the incident light path, comprises a beam splitter (PBS) and a polarizer plate. The polarizing beam splitter is then used to guide the incident light to pass the focus lens on the 3D sinusoidal-groove linear-reflective grating. The light beam reflected from the 3D sinusoidal-groove linear reflective grating is collected by a polarizing beam splitter and the reflected light is deflected through the focus lens to the QPD. 

The prototype sensor has a size of 90(L) × 75(W) × 50(H) mm. As shown in Figure 1, the collimated light, as provided by a laser diode (Thorlab, Model: CPS532-C2) will pass through a polarization beam-splitting device (Lambda, Model: OI-10.0BCQ-442) and enter a focus lens (Thorlab, Model: LB1757) on the stainless steel bar. The stainless steel bar has a three dimensional sinusoidal grating, which is a superposition of periodic sinusoidal waves in the X- and Z-directions with spatial wavelengths of 350 μm and amplitudes of 0.5 μm. The light reflected from the grating will pass through a polarizing beam splitter and, after a focus lens, is detected by the QPD (Ontrak, Model: PSM2-4Q). The reflective-type 3D sinusoidal grating, wherein a profile equation of the reflective-type 3D sinusoidal grating, h(x, z), is expressed as:(1)$$h(x,z)=-A_{x}\sin\left(\frac{2\pi x}{P_{x}}\right)-A_{z}\sin\left(\frac{2\pi z}{P_{z}}\right)$$
where *x* and *z* are, respectively, the positions of the reflective-type 3D sinusoidal grating along X- and Z-axes. In addition, the terms *P_x,z_* and *A_x,z_* correspond to wavelengths and amplitudes of the reflective-type 3D sinusoidal grating. In this study, the *P_x,z_* and the *A_x,z_* are 350 μm and 0.5 μm, respectively. First, as shown in Figure 1, a light beam is projected onto the target surface (T_1_) of the grating. The tilt angle θ_z_ about the Z-axis can be calculated from the moving distances of the optical spot on the QPD by applying the auto-collimation principle as shown below:(2)$$\theta_{X}=\frac{\Delta v}{2f},\theta_{Z}=\frac{\Delta u}{2f}$$
where *u* and *v* are the displacement in X- and Z-axes of the QPD, respectively, and *f* is the focus distance from the lens to the target surface. The second step is measuring of the linear displacement and vertical straightness of the precision linear stage. The light beam is projected onto the target surface (T_2_) of the 3D sinusoidal grating by moving the Z-axis translation stage. The partial differentiation for slope equations of 3D grating, *m_x, z_* (x, z), in X- and the Z-axes are as follows:(3)$$m_{x}\left(x,z\right)=\frac{\partial h\left(x,z\right)}{\partial x}=\frac{2\pi A_{x}}{P_{x}}\sin\left(\frac{2\pi x}{P_{x}}\right)$$
(4)$$m_{z}\left(x,z\right)=\frac{\partial h\left(x,z\right)}{\partial z}=\frac{2\pi A_{z}}{P_{z}}\sin\left(\frac{2\pi z}{P_{z}}\right)$$
where *x* and *z* are the positions of the reflective-type 3D sinusoidal grating along X- and Z-axes, respectively. We can also apply the auto-collimation principle to measure *m_x_(x,z)* and *m_z_(x,z)* from QPD [21]:(5)$$u=\left({tan}^{-1}m_{x}\left(x,z\right)\right)\times 2f$$
(6)$$v=\left({tan}^{-1}m_{z}\left(x,z\right)\right)\times 2f$$

With reference to Equations (5) and (6), QPD receives the sensing signal which represents the position variation of the 3D sinusoidal grating in X- and Z- directions. Therefore, the axial motion displacement volume and the radial displacement volume of the linear stage are calculated using a slope equation and a position signal equation to calculate the sensing signals of the QPD. In other words, the X- and Z-axes motion signals are obtained only as a function of u and v values. However, with reference to Equations (5) and (6), the signal discriminant analysis needs to be made to define the quadrant within which the initial and terminal points of the motion signals lie.

## 3. Analyzing the Sequence Signal by Using the Triangular Wave-Based Pulse Triggering Method

When a pulse is sent into the input, the output will change triggering the flip flop. Indeed, flip flops are often used to design a counter and store the data in multi-bit numbers. Such a counter needs to be connected to multiple flip flops as a sequential circuit. The counter then counts the number of trigger pulses that are applied to the input. A counter circuit is usually constructed on a series of flip flops connected in cascade. When a single pulse is applied to a counter that stores multi-bit data, it causes the bit to move its position.

Counters are widely used in digital circuits, and commonly manufactured as separate integrated circuits or incorporated as parts of larger integrated circuits. Inside each counter block in a circuit an interrupt module monitors signals coming from interrupt sources within the block. Potential interrupt sources are state machines, counters, combinatorial logic, and external signals. When an interrupt source is asserted, it triggers the interrupt module to latch on to the event, thus generating an interrupt (directly or indirectly) in the CPU (Central Processing Unit).

A clock pulse used to operate a flip flop is illustrated in Figure 2a. The pulse goes from a low level (0 volt), the positive logical 0 condition, to a high level (+5 volts), the positive logical 1 condition; it alternates between these two logic levels at a fixed frequency rate. The clock signal shown in Figure 2 has two transitions: (i) the leading edge describes the transition from low to high; and (ii) the clock trailing edge describes the transition from high to low. Flip-flop circuits are triggered either by the leading edge or by the trailing edge, as will be seen later. Two basic types of flip-flop circuits exist: level- and edge-triggered. With reference to Equations (5) and (6), the QPD receives the sensing signals, which represent the variation in the slope signals in X- and Z- directions. We could obtain an initial position and an end position by a position signal equation of the 3D grating and analyze the actual X-axis displacement along with position variance of the moving stage and the straightness error of the linear stage. However, the dynamic measurement accuracy was not only unstable on extraction from Equations (5) and (6), but was also sensitive to vibration from environmental disturbance, on measurement.

By passing through a comparator, the square wave signal output (I_S_) from the QPD is converted, as shown in Figure 3. At the first stage of signal discriminant analysis, we apply a positive edge-triggering counter and then a negative edge-triggering counter, as shown in Figure 3. The edge triggering allows a device to create a very fine level trigger which is faster than all external feedback loops, so that the device can accept inputs quickly, and close off the entrance in time before its changing output changes the input value. In addition, by passing through an integrator, the corresponding signal output (*I_T_*) is transformed into a triangular-wave signal of the *I_S_* signal, which is periodic but not sinusoidal, as shown in Figure 3. The most important feature of a triangular wave is that it has equal rise and fall times, in contrast to a saw-tooth wave which has different rise and fall times. In this work, to get a steady subdivision signal, we have used the active level-triggered method for analyzing sequence triangular-wave signals.

In the second stage of the signal discriminant analysis procedure, we applied active high-level triggering counter as shown in Figure 3. The active high/low level are *I_T_*±*I_SV_* in which *I_SV_* stands for subdivision voltage. We can get the subdivision voltage (*I_SV_*) after the subdivision number (*N_SB_*). The relation between *I_SV_* and *N_SB_* is:(7)$$I_{SV}=\frac{I_{T-MAX}}{N_{SB}}$$
where *I_T-MAX_* is the peak amplitude of the triangular-wave signal (*I_T_*). The period length (*Q_SB_*) of the subdivision signals could be defined by:(8)$$Q_{SB}=\frac{P_{X,Z}}{N_{SB}}$$

Consequently, the active high/low level could vary corresponding to the triangular signal output (*I_T_*). The use of either the higher or lower voltage level to represent either logic state is arbitrary. Active high and active low states can be mixed at will, and the two logical states are usually represented by two different voltages. High and low thresholds are specified for each logic family. When the voltage of the signal output (*I_T_*) reaches the active high/low level, we can get the pulse-train signal as shown in Figure 3. In this way, *O_P_* is equal to the numbers of the period length of subdivision signals. Finally, having determined the *O_P_*, the total displacements *D_X,Z_* along X- and Z-axes can be represented as follows:(9)$$D_{x,z}=P_{x,z}\times O_{P}$$

In this paper, the detection signals of the optical encoder outputs two different sine wave analog signals. Interpolation circuit board, as shown in Figure 3, converts these signals to digital through AD converter, then increases resolution per one signal output. Therefore, the resolution is 0.175 μm using interpolation circuit with *N_SB_* = 2000.

## 4. Experimental Setup

To implement the triangular wave-based signal subdividing algorithm in the encoder system, we need a hardware that possesses at least two sampling channels of high-precision analog-to-digital converters having a high sampling rate and a high-speed digital signal processor (DSP) to compute the total displacements D_X,Z_ along the x- and z-axes. The 3D signal of the QPD is captured using a real-time controller system (National Instruments, Model: cRIO-9039) with 16-bit resolution. The real-time controller system, which is composed of the reconfigurable FPGA chip (Xilinx Kintex-7 325T FPGA) and A&D (analog NI-9223 & digital signal NI-9361) input module, is used to efficiently subdivide the electronic signal for data acquisition and analysis. For a commercial product, the sampling rate of the analog-to-digital converter can reach 1 MHz, so the positional values of the QPD can be obtained in a very short time.

In performing the experiments, the prototype sensor and the 3D sinusoidal-groove linear reflective grating are set using precision linear air-bearing stage (Aerotech, Model: ABL1500) and air-bearing rotary stage (Aerotech, Model: ABRS150MP) on the granite base plates, as shown in Figure 4. The precision linear air-bearing stage coupled with a low thermal expansion linear encoder provides a travel range of 150 mm and a resolution of 1nm. Similarly, the precision air-bearing rotary stage also coupled with a low thermal expansion linear encoder provides a travel range of 360^o^ and a resolution of 0.03 arc second. The thermal expansion coefficient for standard ABL1500 and ABRS150MP stages is 7.5 ppm per °C. 

On the other hand, the 3D sinusoidal linear reflective grating structure on a stainless steel bar was machined by an ultrasonic elliptical vibration cutting system. The 3D micro-structured surface on the middle of the stainless steel bar is as shown in Figure 5, as a result of the superposition of periodic sinusoidal waves along the x- and z-axes (spatial wave lengths and amplitudes are 350 μm and 0.5 μm, respectively). The accuracy of machining along the x- and z-axes was measured to be 0.274 μm and 0.095 μm, respectively. The profiles of the textured grooves were measured by an optical surface profiler (ZYGO NewView6200). The top and bottom surfaces of the stainless steel bar are well-polished with a reflection coefficient of about 0.87 at 532 nm.

To eliminate environmental vibrations that limit performance, all measurements were made in contamination-free cleanrooms, which are certified Class 10,000. The average atmosphere temperature was 20.4 °C ± 0.32 °C with a relative humidity controlled between 40% and 50%. Cleanroom access is controlled by an airlock, which utilizes positive pressure to prevent outside air from entering. The average atmospheric pressure was 761.6 mmHg with a standard deviation of 0.6 mmHg.

To demonstrate the accuracy and the effectiveness of the measurement system, we have performed verification tests for positioning errors using an interferometer measurement system (Renishaw XL-80) with 1 nm resolution by moving the stage along a distance of 100 mm for several runs. The vertical straightness was measured by an interferometer system (Renishaw XL-80) with 0.01 μm resolution. In this study, we made the incident light from a light source mounted on a moving stage to illuminate a 3D sinusoidal-groove linear grating disposed at a guiding rail, in which the reflected light from the 3D sinusoidal-groove linear grating forms a light spot on a QPD, which reacts by generating a plurality of sensing signals. The position difference of the light spot varies with the movement of the stage, so it further alters the voltage of the sensing signals of the QPD. The optical sensor head was mounted on a manual stage unit consisting of a manual translation station that can move along the x-axis and three manual tilt stages that can apply inclination angles to the optical sensor head. By adjusting the manual stages, the optical sensor head can be aligned with the 3D sinusoidal-groove linear grating.

## 5. Experimental Results

The proposed method has been validated experimentally as shown in Figure 6. To measure the tilt angle of the linear stage, the light beam is projected onto the grinding surface of the stainless-steel bar as shown in Figure 6. In regard to the eight given rotation angles of −3°, −2°, −1°, −0.5°, 0.5°, 1°, 2°, and 3°, the tilt angles (θ_Z_) were measured and calculated by substituting the sensing signals from the QPD into Equation (2). The tilt angle tests show that the results of every 10 data points obtained from the proposed system almost coincide with the given rotation angles (Figure6). The maximum error in tilt angle measurement was less than 0.058°. The percentage errors in the tilt angle tests of the eight given rotation angles were found to be 0.73%, 1.19%, 0.74%, 0.49%, 1.45%, 0.64%, 2.94% and 1.48%, respectively. Subsequently, the eight error bars of tilt angle measurement are ±0.189°, ±0.113°, ±0.068°, ±0.031°, ±0.033°, ±0.086°, ±0.103°, and ±0.146°, respectively.

The average and standard deviations of the differences in the measured errors between those from the proposed system and those from the interferometer system were taken and reported as the measurement accuracy and its repeatability. The verification data are shown in Figure 7 and Figure 8, obtained with the stage moving at a fixed speed of 5 mm/s. The 3D sinusoidal-groove linear grating was moved by a precision linear air-bearing stage along the X-axis while the prototype sensor was kept stationary. The extracted signals from the QPD could be inserted into Equation (9) to calculate the vertical straightness and displacement.

In Figure 7, the linear displacement tests show that the results of every 10 data points obtained from the proposed system almost coincide with those obtained from the interferometer system. The percentage errors in the linear displacement tests of the ten given positions were found to be 0.05%, 0.56%, 0.64%, 1.15%, 1.03%, 1.24%, 1.16%, 1.43%, 1.49% and 1.56%, respectively. The maximum error of linear displacement measurement was less than 0.239 μm. Figure 7 also shows the experimental results of sequence linear displacement measurements in the X-axis corresponding to the travel ranges of 100 mm. Furthermore, the ten error bars of linear displacement measurement are ±0.067 μm, ±0.093 μm, ±0.08 μm, ±0.076 μm, ±0.054 μm, ±0.052 μm, ±0.067 μm, ±0.073 μm, ±0.084 μm, and ±0.058 μm, respectively. 

Figure 8 shows the vertical straightness results of every 10 data points of Z-axis corresponding to travel ranges of 100 mm. The data have been taken from the proposed system and the interferometer system along with position variance of the moving stage. The maximum error of the vertical straightness measurement was 0.188 μm. Similarly, Figure 8 shows the experimental results of sequence vertical straightness measurements corresponding to the travel ranges of 100 mm. Also, the ten error bars of vertical straightness measurement are ±0.011 μm, ±0.014 μm, ±0.013 μm, ±0.007 μm, ±0.016 μm, ±0.01 μm, ±0.011 μm, ±0.011 μm, ±0.014 μm, and ±0.013 μm, respectively. As can be seen from the experimental results, the proposed method was able to measure simultaneously the linear displacement and vertical straightness of the precision linear air-bearing stage by using the proposed triangular wave-based signal analysis method.

## 6. Conclusions

Overall, this study proposes a compact and accurate method for a straightforward and reliable means of extracting the tilt angle, linear displacement, and vertical straightness of the precision linear stage. It has been validated by comparing the experimental results of the tilt angle, linear displacement, and vertical straightness error with those obtained using commercial interferometer measurement system and precision rotary stage. Significantly, the proposed system combined 3D sinusoidal-groove linear reflective grating with a novel triangular wave-based sequence signal analysis that gives precise measurements of the tilt angle, linear displacement, and vertical straightness error. Thus, compared to the existing interferometer- or diffraction-based methods, the arrangement is similar to that of a linear encoder and can make online measurements of stage errors for the analysis of automatic processes as well as for real-time monitoring. It should be noted that the tilt motion measurement is captured and extracts the analog signal from QPD. Thus, measurement accuracy and its repeatability of tilt motions are not as good as the experimental results of the linear displacement and vertical straightness. Additionally, the fabrication is easier to miniaturize due to the straightforward and compact design which can be applied to the multi-axis system in the future study. The dual-beam laser spots could be applied for online tilt motion, straightness, and position measurement of the precision linear stages. Moreover, additional modules for error compensation can be added for most industrial applications.

## Figures and Tables

**Figure 1 sensors-19-02816-f001:**
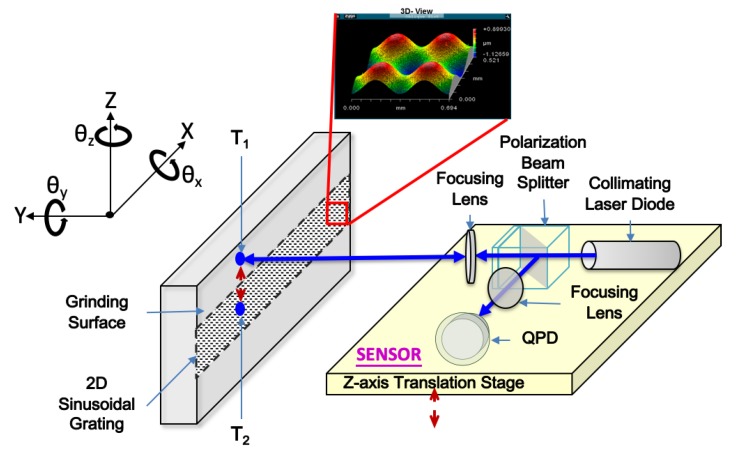
Configuration of the proposed measurement system.

**Figure 2 sensors-19-02816-f002:**
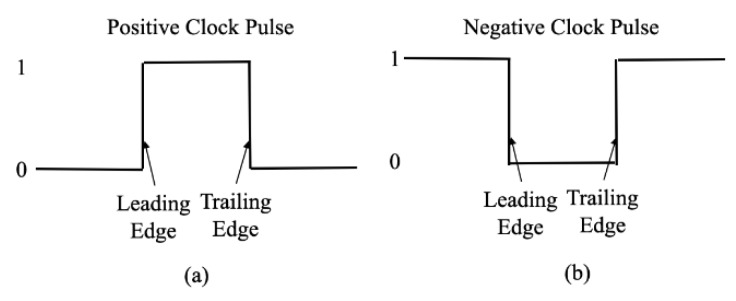
Clock Waveform: (**a**) Positive Clock Pulse; (**b**) Negative Clock Pulse.

**Figure 3 sensors-19-02816-f003:**
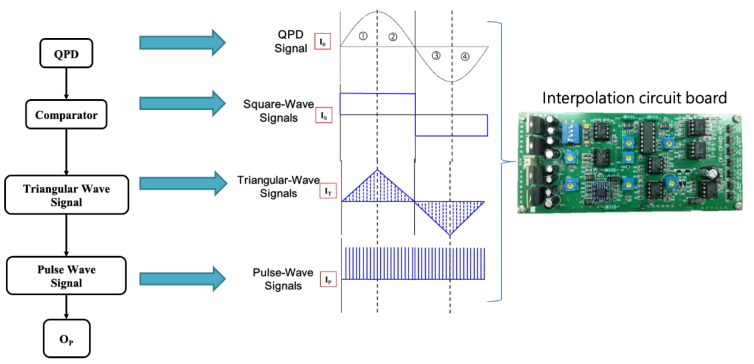
Photograph of the circuit board of the proposed triangular wave-based pulse-triggering method, and scheme illustration of the corresponding signal output in measurement system. *I_0_*: quadrant photo diode (QPD) output signal, *I_S_*: the corresponding signal output from the QPD, *I_T_*: the triangular signal output of the *I_S_* signal, and *I_P_*: the pulse signal output of the *I_S_* signal. The *O_P_* is the number of the pulse wave edge triggering counting.

**Figure 4 sensors-19-02816-f004:**
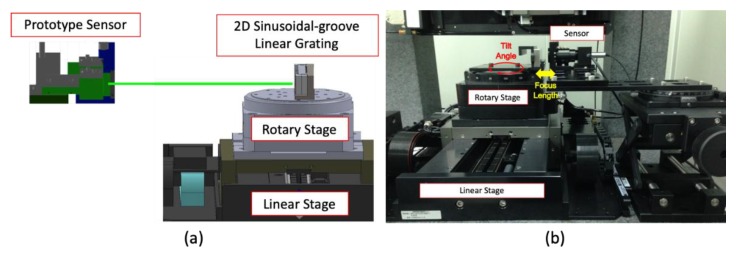
(**a**) Schematic of experimental setup with the proposed system. (**b**) Photograph of the experimental system.

**Figure 5 sensors-19-02816-f005:**
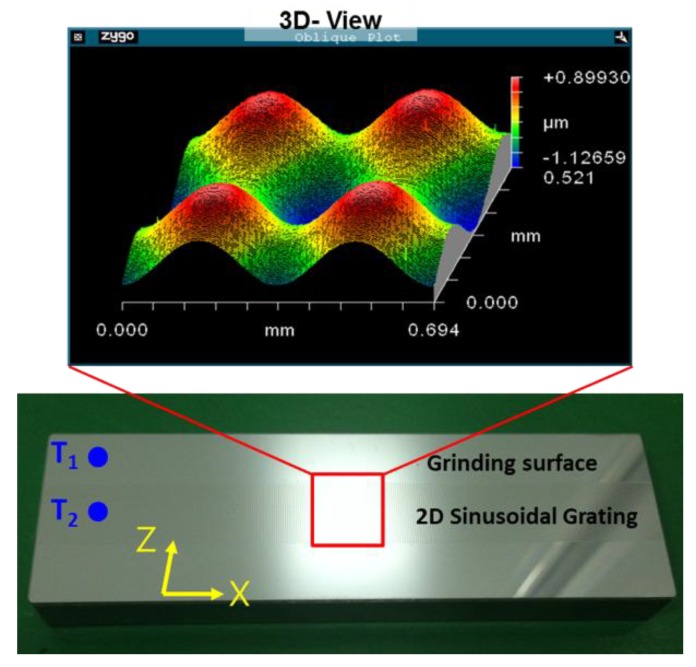
The measured profiles and photograph of the two-dimensional sinusoidal grating.

**Figure 6 sensors-19-02816-f006:**
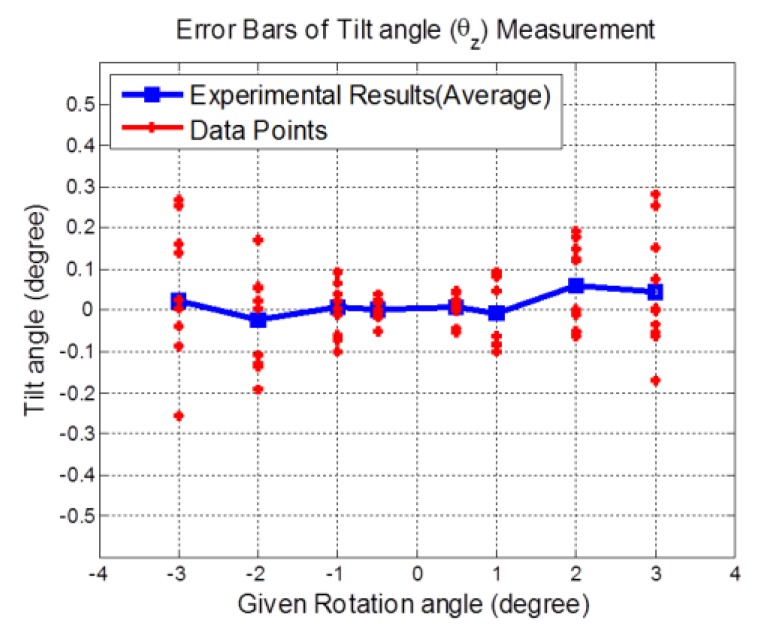
The measured experimental results of tilt angle (θ_Z_) tests by the proposed system.

**Figure 7 sensors-19-02816-f007:**
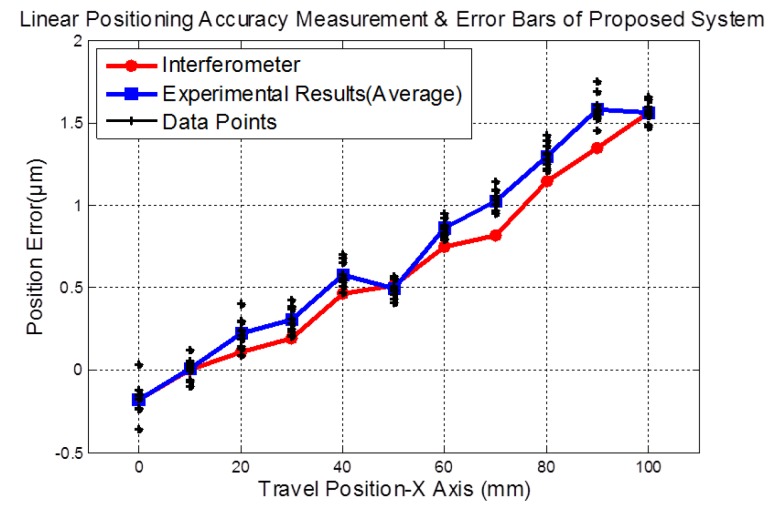
The measured average results and error bars of linear positioning by the developed system and an interferometer measurement system (Renishaw XL-80).

**Figure 8 sensors-19-02816-f008:**
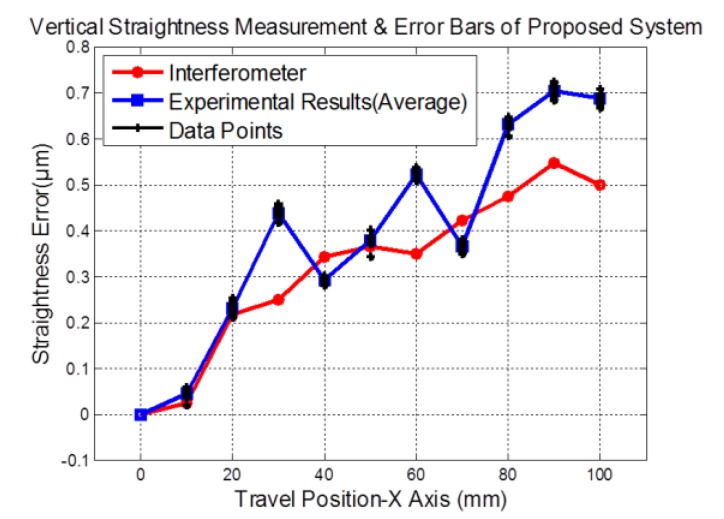
The measured results and error bars of vertical straightness by the developed system and an interferometer measurement system (Renishaw XL-80).

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
