# Peer review of "An Approach to Measure Tilt Motion, Straightness and Position of Precision Linear Stage with a 3D Sinusoidal-Groove Linear Reflective Grating and Triangular Wave-Based Subdivision Method"

_sensors, 2019, doi:10.3390/s19122816_

Round 1
Reviewer 1 Report
The paper by H-A Tsai and Y-L Lo entitled “An approach for measuring tilt motion, straightness and position of precision linear stage with a hybrid 2D Sinusoidal-Groove Linear Reflective Grating and Triangular wave-based Subdivision method” presents a novel compact method for simultaneously measuring linear displacement and vertical straightness error of a moving linear air-bearing stage based on utilization of the hybrid 2D sinusoidal-groove linear reflective grating and a novel triangular wave-based sequence signal analysis approach.
The topic of the paper is important from the technical point of view, and the methods used seem to be adequate. However, the paper has many drawbacks, which I summarize below.
Line 100 – “… the proposed measurement system”. In this paragraph, there is no good correspondence of the Fig. 1 content to its description, e.g. “collimating lens” – “focusing lens” etc., thus making the whole paragraph unclear. The end of the paragraph (from line 108) should be removed.
Starting from the above paragraph, the term “hybrid 2D sinusoidal-groove linear reflective grating” is used. My understanding is that it is actually 3D grating on the side surface of metal bar, as later follows from Fig. 7, thus the word “hybrid” is a bit misleading.
Probably, some representation of the grating pattern can be given in Fig. 1.
L115: In the operation description, I think the present indefinite form is more suitable.
L119: Sinusoidal array -> grating
L120: It is better to use the same units such as um, so 500nm -> 0,5um. Sample -> grating
L121: come -> comes
Eq. 1: It is worth to assign names to the variable h(x,z)
Eq. 2: Variable v is not present
L135: It is worth to assign names to the variables m(x,z)
L147: The sentence to be corrected
L154: each block -> each counter block
Figure 2: Right part seems to be incorrect, “0” and “1” labels to be exchanged, if we talk about logical levels
L169-183. This description seems to be excessive.
L190: The meaning of “I_s” – is it electric current (similar for I_T, I_sv, etc)? was -> is
L192: applied -> apply
L190-200: The stage by stage signal conversion description is not clear.
L205-206: The sentence should be corrected.
Figure 4. The difference between it and Fig. 3 is not clear.
Figure 5 – has no meaning, as it is just a photo of a hardware, not an electric circuit.
L252, paragraph, and Figure 7. Here “3D structure” term appears, which seems better describes the grating structure (see above comment L100).
L283: is experimentally demonstrated
L284: top surface – does it actually mean “side surface” of the metal bar?
L289,290, and further: I believe the precision of the measurements (relative error) should be also given as a percentage of the displacement value
Figure 8. The difference between it and Fig. 6 is not clear. They look the same but with different “zoom” value.
Figure 10. The difference between it and Fig. 11 is not described clearly. It seems both figures should have the same axis scales and limits (and Figs 12, 13 as well).
Figs. 10 – 13: Vertical label units (um?) are not shown correctly, at least in my PDF file. Again, it is worth to indicate percentage of relative error.
L331, conclusions. I cannot judge about the compactness and accuracy of the proposed system as no comparison with other systems is given.
In the conclusion I would recommend the paper for publication upon significant (major) improvements and clarifications, including English language.
Author Response
We would like to express our gratitude to you for your consideration of our manuscript entitled, “An approach to measure tilt motion, straightness and position of precision linear stage with a 3D Sinusoidal-Groove Linear Reflective Grating and Triangular wave-based Subdivision method” for publication in Sensors.
The content of this article has been revised by the reviewer’s comments and suggestions. And the grammar of this article has been modified by an English native speaker. All the changes made in the manuscript have been highlighted in red. We look forward to hearing from you regarding your final decision on our manuscript.

Reviewer 2 Report
This paper focus the measurement of the straightness, which is one of the most important geometric error source in the precise displacement. Compared to the method before, the proposed method can simultaneously measure tilt angle, linear displacement, and vertical straightness error. The paper is well organized and the results are clear. However, I think the paper should be made a careful revision before its publication in Sensors.
(1) Please delete “he” in the sentence “good as the he experimental results”, in Line 343.
(2) What is the meaning from the Line 108 to Line112? Should they be deleted?
(3) It should be “a” rather than “an” in the first senRegarding to the measurement principle and results, I have the following concerns.
(4) Line 117-118 said the focus lens was on the stainless steel bar, and also said the 2D array was on the stainless steel bar. Where is the “stainless steel bar” in Fig.1, left or right?
(5) Authors mentioned the parameter “v” in line 130, however, there is no parameter “v” in Equation (2). Since the angle measurement depends on the Equation (2) directly, this mistake is serious.
(6) Can you give the maximum measurable range and influence factors of your sensor for the tilt angle, linear displacement, and vertical straightness measurement, respectively?
(7) Can you give the possible reason that why the results in Fig.10-13 show the increasing tendency with the travel position? Does some “system error” exist in the displacement system?
Some other mistakes should be corrected:
(8) Unit of the position error in Figure 10-13 need to be corrected.tence ”an pulse is sent into” in Line 147.
(9) It should delete “in” in Line 310.
Author Response
We would like to express our gratitude to you for your consideration of our manuscript entitled, “An approach to measure tilt motion, straightness and position of precision linear stage with a 3D Sinusoidal-Groove Linear Reflective Grating and Triangular wave-based Subdivision method” for publication in Sensors.
The content of this article has been revised by reviewer’s comments and suggestions. And the grammar of this article has been modified by an English native speaker. All the changes made in the manuscript have been highlighted in red. We look forward to hearing from you regarding your final decision on our manuscript.
